**∂ | Open Peer Review** | Evolution | Observation

# An early diverging SQR enzyme in Antarctic *Gloeobacterales* indicates sulfide tolerance in thylakoid-lacking cyanobacteria

Louise Hambücken,[1] Edi Sudianto,[1] Elie Verleyen,[2] Jimmy H. Saw,[3] Denis Baurain,[1] Luc Cornet[1]

**ABSTRACT** Oxygenic photosynthesis, which converts solar energy into carbohydrates via a linear electron transport chain and two photosystems (PSII and PSI), first appeared in cyanobacteria approximately 3.3 Ga and drove the Great Oxidation Event around 2.4 Ga. During this period, euxinic conditions—characterized by sulfidic, anoxic oceans —posed a metabolic challenge to cyanobacteria, as sulfide inhibits PSII, the reaction center responsible for water splitting. Here, we report the presence of a sulfide-quinone reductase (SQR) enzyme in Antarctic representatives of *Gloeobacterales*, the earliest-branching cyanobacterial lineage. Phylogenetic analyses consistently position these SQR sequences at the base of the cyanobacterial clade, likely predating the multiple lateral transfers reported for this gene in the phylum. Additional searches in metagenomic data sets indicate that such sequences are restricted to cold environments. Our findings unveil possible adaptive strategies of early cyanobacteria to cope with sulfidic stress and point to Antarctic lakes as preserved natural laboratories for investigating cyanobacterial diversification and the evolution of oxygenic photosynthesis under euxinic conditions.

**IMPORTANCE** The diversification of cyanobacteria during and after the Great Oxidation Event occurred in early Proterozoic oceans that were partially euxinic (anoxic and sulfidic), a condition generally considered incompatible with oxygenic photosynthesis due to photosystem II inhibition. The presence of a sulfide quinone reductase in an early diverging cyanobacterium lacking thylakoids, isolated from Antarctica, suggests that oxygenic and anoxygenic photosynthesis coexisted early on in cyanobacterial evolution. The occurrence of these organisms in Antarctic lakes under euxinic conditions offers a natural laboratory for studying the physiology and adaptation of the first oxygenic photosynthetic organisms.

**KEYWORDS** sulfide quinone reductase, photosynthesis, cyanobacteria, *Gloeobacter*, evolution

Oxygenic photosynthesis is a biochemical process that converts solar energy into carbohydrates by transferring electrons from water through a linear electron transport (LET) chain involving two photosystems (PSII and PSI). This process first appeared in cyanobacteria approximately 3.3 Ga (1), releasing oxygen into both the oceans and the atmosphere as a byproduct and ultimately leading to the Great Oxidation Event around 2.4 billion years ago (2). When oxygen accumulated in the atmosphere, it created oxidizing conditions that promoted the weathering of terrestrial sulfides, as evidenced by the presence of evaporites (sulfate-rich rocks) dated between 2.7 and 2.2 Ga (3). This weathering released large quantities of sulfate into the oceans, which could then be used by sulfate-reducing bacteria, producing sulfide in the process (4). The accumulation of sulfide transformed the oceans into an anoxic and sulfidic environment—known as the euxinic ocean—as proposed by Canfield (5). The $Fe^{2+}$ present in the oceans, which had first reacted with oxygen to form ferric hydroxides

Address correspondence to Luc Cornet, luc.cornet@uliege.be.

The authors declare no conflict of interest.

See the funding table on p. 6.

and banded iron formations characteristic of oceanic oxygenation, began instead to react with sulfur, forming pyrite ($FeS_2$) (4). According to Canfield, these euxinic conditions were widespread during the early and mid-Proterozoic (5), though subsequent studies tempered this view (4). Johnston et al. demonstrated that $H_2S$ was nevertheless sufficiently abundant in the oceans for it to pass the chemocline and to reach the surface, impacting the entire water column and releasing $H_2S$ into the atmosphere at the end of the first era of the Proterozoic: the Paleoproterozoic (6). As a result, cyanobacteria living in the photic zone of the mid-Proterozoic experienced alternating oxygenated and sulfidic conditions (7). In this biogeochemical context, the diversification of oxygenic cyanobacteria may appear paradoxical as sulfide is known to inhibit photosystem II and thus oxygen release, by an as-yet-unclear mechanism (8, 9).

In most cyanobacteria, oxygenic photosynthesis takes place in specialized membrane compartments known as thylakoids. However, the earliest-diverging cyanobacterial order, *Gloeobacterales*, lacks thylakoids, and its LET chain is instead located in the cytoplasmic membrane, where oxygenic photosynthesis occurs (10). Given the ancestral nature of *Gloeobacterales*, the emergence of oxygenic photosynthesis must have taken place within the cytoplasmic membrane before the relocation of the LET chain to the thylakoids (11). *Gloeobacterales* have been identified in a variety of environments, including cold, wet-rock, and low-light environments (12). Within their known diversity, there exists a group of taxa represented exclusively by metagenome-assembled genomes (MAGs) that are endemic to Antarctica.

During the Proterozoic, terrabacterial anoxygenic photosynthesis was widespread (13). According to Nishihara et al. (13), anoxygenic photosynthesis is thought to have preceded oxygenic photosynthesis, but see references 14, 15 for different views in the longstanding debate on the emergence of the latter. Anoxygenic photosynthesis can notably involve a LET chain where sulfide-quinone reductase (SQR) catalyzes $H_2S$ oxidation and supplies the resulting electrons to PSI (11, 16, 17). Alongside purple sulfur bacteria (e.g., *Chromatium*) and green sulfur bacteria (e.g., *Chlorobi*), for which sulfide oxidation is the sole photosynthetic pathway, some cyanobacteria are also capable of using the SQR enzyme intermittently (18, 19). In this study, we report the presence of an early diverging form of the SQR enzyme in Antarctic representatives of the *Gloeobacterales*.

We built a hidden Markov model profile of the *Gloeobacterales*-specific SQR type I (SQR-I) using the protein sequences derived from three Antarctic *Gloeobacterales* MAGs (GCA_949127895.1, GCA_949127685.1, and GCA_038245785.1) and performed an orthology search against 107,237 representative bacterial species genomes from the GTDB database (20) (Fig. S2). This search yielded 19,081 homologous sequences, from which we computed an unrooted multigenic family tree (Fig. S2) where the SQR-I sequences of cyanobacteria all form a monophyletic group. Interestingly, the three Antarctic *Gloeobacterales* SQR-I sequences occupy the basal position of this group, which is consistent with the position of *Gloeobacterales* in phylogenomic trees and suggests that these sequences are the earliest diverging SQR-I sequences in cyanobacteria. Owing to its position, we used this Antarctic clade to root a smaller tree inferred from the 192 cyanobacterial SQR-I sequences (Fig. 1). Although cyanobacteria can possess both SQR-I and II (e.g., *Synechocystis* sp. PCC 6803 [18]), we identified only SQR-I in the *Gloeobacterales* MAGs. Previous studies have reported lateral gene transfers (LGTs) among cyanobacteria, notably involving *Thermostichales* (21), the second cyanobacterial order to diverge within cyanobacteria. In our smaller tree, several well-recognized cyanobacterial orders of Strunecký et al. (22) (e.g., *Oscillatoriales* and *Nostocales*) are exploded and intermingled with strong bootstrap support, indeed evidencing pervasive LGT in SQR-I evolution.

The SQR sequences reported here are closely related but exhibit a common long branch. Consequently, we assessed their placement using several evolutionary models (see Supplemental Materials) during phylogenetic inference, all of which consistently recovered this early diverging position. However, we cannot completely rule out a potential long-branch attraction artifact (23) that would place these sequences at the

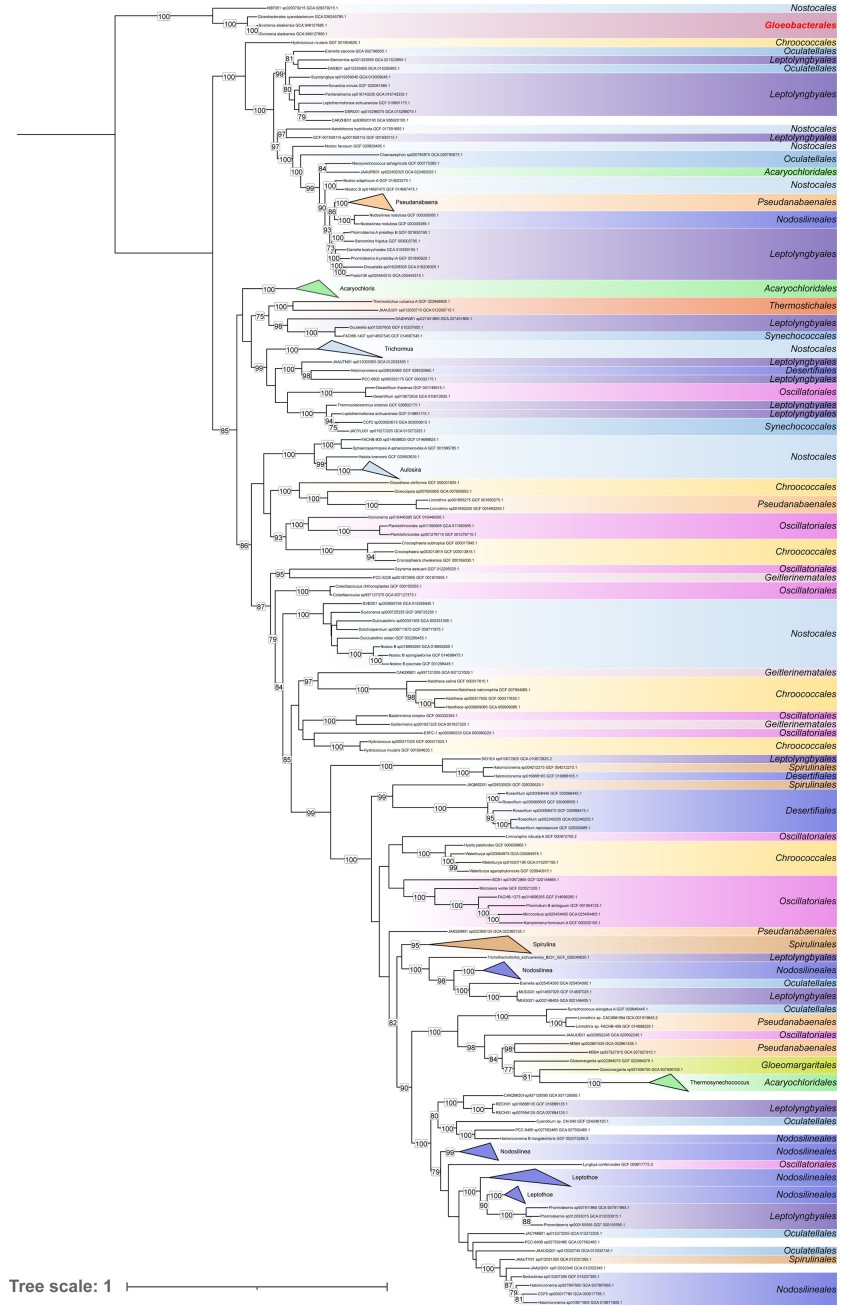

**FIG 1** Phylogenetic tree of cyanobacteria SQR-I sequences. The tree, constructed from 192 sequences across 438 unambiguously aligned positions, was inferred using IQ-TREE with the best-fit model (Q.PFAM + I + R5). It is rooted on the *Gloeobacterales* sequences, based on the topology of the multigenic family tree in Fig. S2. The classification proposed by Strunecký et al. (22) was applied to the genomes using the same color code as in that study. The first appearance of a Strunecký et al. (22) order is indicated on the tree. Only bootstrap proportions above 70% are shown.

base of the cyanobacterial SQR-I tree, despite them corresponding to more recent but highly divergent sequences. In that case, an alternative interpretation would be that they result from a lateral gene transfer into a subgroup of "cold" *Gloeobacterales*. To find such a potential gene donor, whether within or outside cyanobacteria, we performed extensive orthology searches (Fig. S1), which only returned a single additional, closely related sequence. This sequence originates from a *Rhizonema* MAG (GCA_029379215.1) recovered from a cyanolichen inhabiting a cold alpine environment at 1,600 m elevation

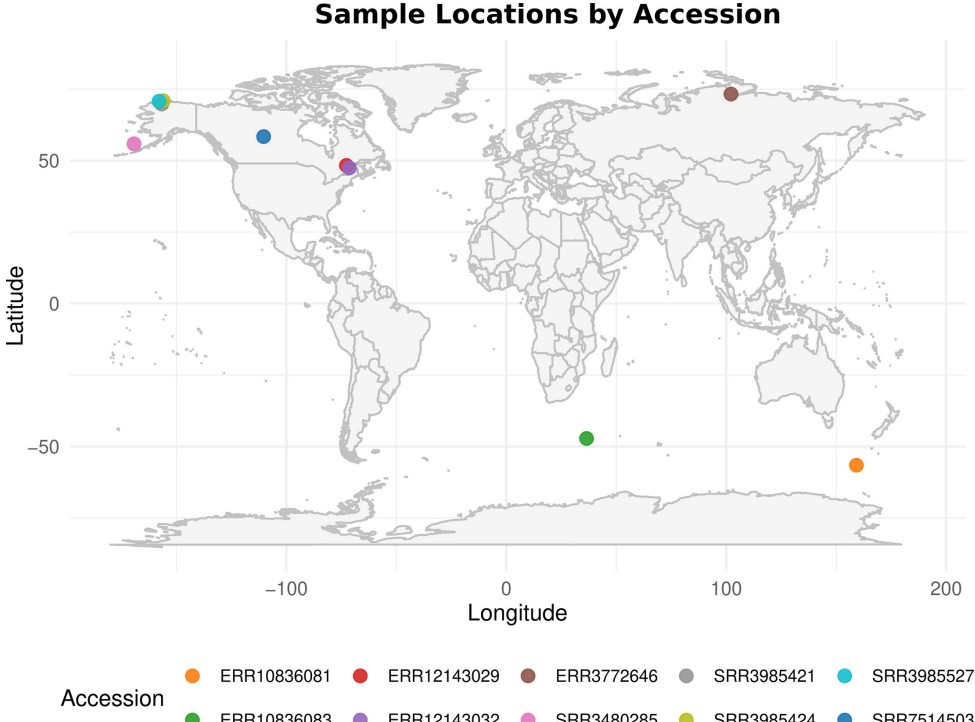

**FIG 2** Geographic localization of the SRA data sets identified by Logan.

(M. Dal Forno, Botanical Research Institute of Texas, personal communication). We then performed a nucleotide kmer-based search using SQR-I sequences from *Gloeobacterales* as queries to detect kmers specific to the *Gloeobacterales*-type SQR-I across the largest DNA sequencing data set, the Sequence Read Archive of the National Center for Biotechnology Information. This analysis returned 11 matches, mostly (9/11) originating from cold environments (subarctic, subantarctic, and polar) (Fig. 2; Table S1). The SQR-I sequences from these SRAs could not be fully assembled into metagenomes or binned into MAGs; only partial hits were recovered. Nevertheless, the specificity of the kmers used by Logan (24) ensures the presence of these sequences within the SRAs and indicates that Antarctic lakes, as well as similar cold environments, are another suitable habitat for *Gloeobacterales* (25, 26), opening new avenues for the study of these thylakoid-less cyanobacteria.

In conclusion, although we cannot exclude that cold-related environmental factors may play a role in these results, the SQR-I sequences of Antarctic *Gloeobacterales* MAGs reasonably appear to be the earliest cyanobacterial SQR-I reported to date, having diverged from the ancestral sequence to adapt to cold environments. Some Antarctic lakes are meromictic, exhibiting permanent water column stratification due to strong chemical gradients (e.g., see references 27 and 28). In Ace Lake (Vestfold Hills) and Lake Fryxell (McMurdo Dry Valleys), which are among the best-studied meromictic lakes in Antarctica, the limited water mixing results in anoxic conditions and the accumulation of sulfide in the monimolimnion (bottom water mass), thereby exposing the benthic microbial mats in the deeper parts of these lakes to euxinic conditions. The lakes from which the three MAGs originated are not formally classified as meromictic. However, localized sulfidic zones may still occur, as the presence or absence of sulfide is not systematically assessed or reported during sampling campaigns, and certainly not within the microbial mats in which these cyanobacteria thrive. Therefore, sulfidic conditions may occur in these systems. Although the precise conditions in these lakes differ from those of the Proterozoic oceans, the presence of cyanobacteria carrying an ancestral version of the LET chain alongside an early diverging SQR-I in these euxinic conditions

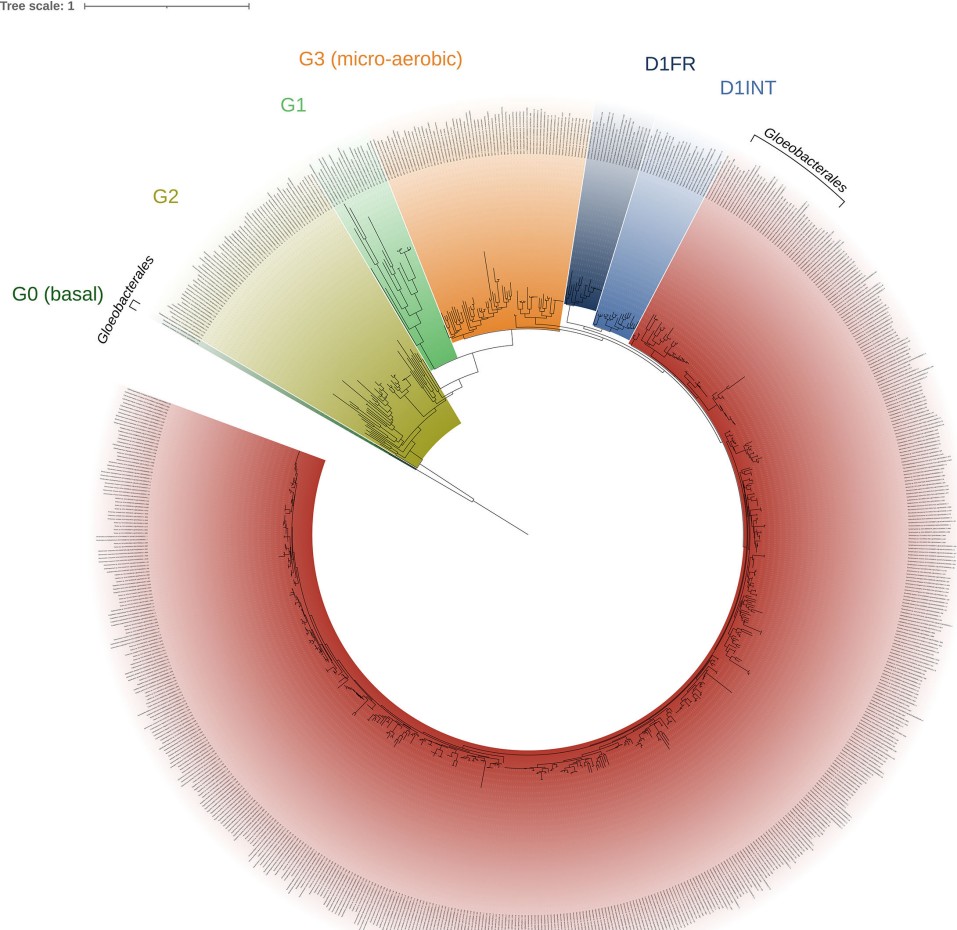

**FIG 3** Phylogenetic trees of the PSII subunit D1 within cyanobacteria. The tree was constructed from 726 sequences aligned over 360 unambiguously conserved positions and inferred using IQ-TREE v3.0.0 under the best-fit model (LG + R9). It was rooted on the *Gloeobacterales* G0 variant, which is proposed to represent the most ancestral form of D1 (30). D1 variants were annotated based on their highest similarity to consensus sequences representing established D1 variant groups retrieved from reference 31. Only bootstrap proportions above 70% are shown.

suggests that meromictic saline or brackish lakes in Antarctica may represent valuable natural systems for studying cyanobacterial diversification under Proterozoic conditions.

Importantly, the SQR-I sequences reported in this study possess both the sulfur-binding sites and the FAD/NADP-binding domains required to catalyze sulfur transfer and reduction processes using FAD/NADP as cofactors. Such an observation implies a probable role in anoxygenic photosynthesis, although this cannot be confirmed without direct physiological demonstration (Fig. S3). While cyanobacteria are known to possess sulfide tolerance mechanisms—such as the microaerobic D1 variant, which permits partial oxygenic photosynthesis under microaerobic conditions (29)—this variant was not identified in any of the *Gloeobacterales* genomes analyzed in this study (Fig. 3).

Since *Gloeobacterales* lack thylakoids, their periplasmic space is considered homologous to the thylakoid lumen (32). Consequently, *Gloeobacterales* SQR are very likely the only SQR enzymes active within the periplasmic space, thus mimicking the ancestral cyanobacterial LET chain subjected to the euxinic conditions of the Proterozoic. Indeed, the presence of an SQR enzyme must allow these organisms to continue generating energy via photosynthesis in the absence of a functional PSII, as the earliest cyanobacteria likely did.

## ACKNOWLEDGMENTS

We thank Manuel Dal Forno from the Botanical Research Institute of Texas for providing information regarding the *Rhizonema* genome.

This work was supported by a research grant (PDR T.0018.24 OR-OX-PHOT-IN-CYN) from the Belgian National Fund for Scientific Research (Fonds de la Recherche Scientifique [F.R.S.-FNRS]) to D.B. L.C. is supported by a mandate from the Belgian National Fund for Scientific Research (F.R.S.-FNRS). L.H. is a FRIA grantee of the Fonds de la Recherche Scientifique – F.R.S.-FNRS. Computational resources were provided by the Consortium des Équipements de Calcul Intensif, funded by the F.R.S.-FNRS under grant no. 2.5020.11 and by the Walloon Region.

L.H. performed all analyses, except for the Logan search. E.S. constructed the GTDB data set and performed the Logan search. L.H., E.S., J.H.S., D.B., and L.C. wrote the manuscript. All authors read and approved the final version of the manuscript.

## AUTHOR AFFILIATIONS

[1]InBioS–PhytoSYSTEMS, Eukaryotic Phylogenomics, University of Liège, Liège, Belgium
[2]Protistology and Aquatic Ecology, Ghent University, Ghent, Belgium
[3]Department of Biological Sciences, The George Washington University, Washington, DC, USA

## AUTHOR ORCIDs

Edi Sudianto https://orcid.org/0000-0002-0771-0385
Luc Cornet http://orcid.org/0000-0002-3420-4488

## FUNDING

| Funder | Grant(s) | Author(s) |
| --- | --- | --- |
| FRS-FNRS | PDR T.0018.24 | Denis Baurain |
| FRS-FNRS | | Luc Cornet |
| FRS-FNRS | FRIA | Louise Hambücken |
| FRS-FNRS | 2.5020.11 | Denis Baurain |

## AUTHOR CONTRIBUTIONS

Louise Hambücken, Conceptualization, Data curation, Formal analysis, Investigation, Methodology, Visualization, Writing – original draft | Edi Sudianto, Formal analysis, Investigation, Methodology, Validation, Writing – original draft | Elie Verleyen, Investigation, Writing – review and editing | Jimmy H. Saw, Writing – original draft | Denis Baurain, Conceptualization, Funding acquisition, Methodology, Writing – original draft | Luc Cornet, Conceptualization, Funding acquisition, Methodology, Project administration, Supervision, Validation, Writing – original draft

## DATA AVAILABILITY

Supplemental materials and methods and additional phylogenetic trees are provided as supplemental material accompanying this article. All data generated and analyzed in this study have been deposited in Figshare and are accessible via this link: https://doi.org/10.6084/m9.figshare.31333204. The repository includes the complete data set, alignment files, phylogenetic trees (Newick format), and associated analysis files.

## ADDITIONAL FILES

The following material is available online.

## Supplemental Material

**Supplemental materials (Spectrum00423-26-s0001.docx).** Supplemental methods, figures, and tables.

## Open Peer Review

**PEER REVIEW HISTORY (review-history.pdf).** An accounting of the reviewer comments and feedback.

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
