## [Reviewer comments · Microbiology Spectrum]

Microbiology Spectrum

An Early-Diverging SQR Enzyme in Antarctic Gloeobacterales Indicates Sulfide Tolerance in Thylakoid- Lacking Cyanobacteria

Louise Hambücken, Edi Sudianto, Elie Verleyen, Jimmy Saw, Baurain Denis, and Luc Cornet

Corresponding Author(s): Luc Cornet, Eukaryotic Phylogenomics, InBioS-PhytoSYSTEMS,

Review Timeline:

Submission Date:	February 26, 2026
Editorial Decision:	February 27, 2026
Revision Received:	March 3, 2026
Accepted:	March 6, 2026

Editor: Blaire Steven

Reviewer(s): The reviewers have opted to remain anonymous.

Transaction Report:

DOI: <https://doi.org/10.1128/spectrum.00423-26>

Re: Spectrum00423-26 (**An Early-Diverging SQR Enzyme in Antarctic Gloeobacterales Indicates Sulfide Tolerance in Thylakoid-Lacking Cyanobacteria**)

Dear Dr. Luc Cornet:

Thank you for the privilege of reviewing your work. Below you will find my comments, instructions from the Spectrum editorial office, and the reviewer comments.

After having read the manuscript and response to previous reviews, I feel that the reviewers concerns have largely been addressed, but I still have some remaining concerns.

As the authors state "the primary objective of this study is to demonstrate that members of Gloeobacterales do, in fact, encode an SQR contrary to previous assumptions that this lineage lacked the enzyme". However the interpretation of these findings surpasses this finding to encompass the early evolutionary divergence of these genes and the function and physiological consequence of the possession of these genes, which I do not feel are well supported by the current data.

I am willing to consider a modified version with a narrower focus on the identification of these genes.

Revision Guidelines

Sincerely,
Blair Steven
Editor
Microbiology Spectrum

Reviewer #1 (Comments to the Author (Required)):

Comment 1: The observation of Gloeobacterales in cold environments is not new (as noted in the paper) and the environments cited do not contain sulfide so the premise of the main interpretation of the paper is not well-supported or a new finding.

We agree that the occurrence of Gloeobacterales in cold environments has been previously reported, as noted in the manuscript. We also acknowledge that the environments from which the Gloeobacterales harboring SQR sequences were retrieved are not described as meromictic or characterized by euxinic conditions. Nevertheless, we would like to note that sulfide concentrations are rarely measured in these environments, and therefore the absence of reported sulfide does not necessarily demonstrate its complete absence. The manuscript has been revised accordingly (lines 181 to 199 of the tracked changes document) to clarify this point. We have also benefited from the expertise of our new co-author, Elie Verleyen, an expert in Antarctic environments, who helped us refine the environmental context.

Comment 2: It is also false that no oxygenic photosynthesis happens in the presence of sulfide. There is evidence of cyanobacterial oxygenic photosynthesis in many sulfidic environments - emerging cave streams, hot springs, etc. Therefore, it is inaccurate to state unequivocally that euxinic conditions inhibit oxygenic photosynthesis.

It is indeed well established that some cyanobacteria have evolved sulfide tolerance mechanisms, including a specific D1 variant that enables partial oxygenic photosynthesis under microaerobic conditions (D1-G3) (Lumian et al., 2021). To assess whether Gloeobacterales possess this particular variant, we conducted a targeted analysis but did not detect it in any of the examined Gloeobacterales genomes. The corresponding figure, along with a description of the associated methodology, has been added to the manuscript (see figure below Figure 3). This finding is consistent with previous studies on this subject (Sheridan et al., 2020; Saw et al., 2021). While the presence of sulfide may not lead to inhibition of oxygenic photosynthesis in cyanobacteria harboring this specialized D1 variant, its absence in Gloeobacterales suggests that sulfide exposure would likely result in inhibition of oxygenic photosynthesis in this group. The manuscript has been revised accordingly (lines 200 to 209 of the tracked changes document).

Phylogenetic trees of the PSII subunit D1 within Cyanobacteria. The tree was constructed from 726 sequences aligned over 360 unambiguously conserved positions and inferred using IQ-TREE v3.0.0 under the best-fit model (LG+R9). It was rooted on the *Gloeobacterales* G0 variant, which is proposed to represent the most ancestral form of D1 (Cardona et al., 2015). D1 variants were annotated based on their highest similarity to consensus sequences representing established D1 variant groups retrieved from Sheridan et al., 2020. Bootstrap values $\geq 70\%$ are shown.

Comment 3: There is very little evidence (presented and previously published) for abundant SQR-encoding *Gloeobacterales* in sulfidic and euxinic environments. Lake Vanda does not have sulfide, where the Aurora genome is from, and the alaskensis are likely from non-sulfidic environments. The authors also do not acknowledge that Lake Fryxall does have sulfide but *Gloeobacterales* have not been recovered from Fryxall. One reference is provided for "In many cases, Antarctic lakes are meromictic and exhibit strong water column stratification, notably due to prolonged ice cover" while ignoring that lack of current evidence for *Gloeobacterales* in those settings.

We have revised the manuscript to provide a more accurate environmental context, notably through the inclusion of an Antarctic expert among our co-authors (lines 181 to 199 of the tracked changes document). The three identified MAGs indeed do not originate from lakes classified as meromictic; however, because sulfide concentrations are rarely measured during sampling campaigns, it is not possible to exclude the presence of localized sulfidic

zones. In any case, our identification, in these basal SQR sequences, of both sulfur-binding sites and FAD/NADP-binding domains, very likely implicate them in anoxygenic photosynthesis.

The absence of Gloeobacterales in Lake Fryxell is not necessarily significant. Indeed, Gloeobacterales are extremely rare cyanobacteria that grow only under very specific conditions, often characterized by low light availability. Although Lake Fryxell contains sulfide, it may not provide the particular environmental conditions required for the growth of these basal cyanobacteria; alternatively, current sampling efforts may simply not yet have detected them.

Comment 4: The manuscript is severely under-cited. The authors provide reference to a perspective piece on the role of SQR in anoxygenic photosynthesis in cyanobacteria rather than citing data that have indicated a role for SQR in this activity. Similarly, there has been decades of debate on the first form of photosynthesis and one paper is cited.

We thank the reviewer for this comment. Our intention in choosing a perspective format was to remain concise while still providing a sufficiently comprehensive overview. We have now added two data-driven references supporting the role of SQR in anoxygenic photosynthesis in cyanobacteria (line 134 of the tracked changes document). Nishihara was indeed the only reference we originally cited for oxygenic photosynthesis emergence, as it is the most recent contribution to this long-standing debate and the one that relies most extensively on genomic data. Again, we have included two additional references to direct readers more broadly to this prolific research domain (line 137 of the tracked changes document).

Comment 5: The SQR-I was recovered from Gloeobacterales in cold environments which are the only genomes included in the study. Concluding from this that SQR-I evolved for some cold environment purpose is not well-supported from the data. SQR-I could have diverged for any number of environmental reasons - what specific evidence other than being in these 3 MAGs could you provide to analyze this in a robust way?

We would like to clarify that the Gloeobacterales recovered from cold environments are not the only genomes included in our study. Our dataset comprised all Gloeobacterales genomes available in GTDB release r220 (Parks et al., 2021), and SQR-I was detected only in those lineages originating from cold environments. To further investigate this pattern, we performed an analysis on 27 million SRA accessions from LOGAN (Chikhi et al., 2024), which encompass nearly all genomes and metagenomes publicly available. This analysis recovered reads specifically affiliated with this SQR-I type predominantly in samples from cold environments, which strengthens the association reported in our manuscript. However, the text has been revised to clarify that other cold-related environmental factors, yet unknown, may play a role in these results (lines 181 and 184 of the tracked changes document).

Comment 6: Lines 116-123: See comments above. Gloeobacterales, including the genomes and the Sqr you analyze here, are in non-sulfidic environments.

This section of the manuscript has been considerably revised to take the comments into account; see previous responses to comments 1 and 3.

Comment 7: Lines 130-132: There is no evidence that these specific taxa are performing anoxygenic photosynthesis. SQR has many roles across diverse organisms. There is no way to conclude from a handful of DNA sequences that this activity is happening in situ.

Our interpretation is based on the known functional roles of SQR in Cyanobacteria and other organisms (Arieli et al., 1994; Miller & Bebout, 2004; Klatt et al., 2015). SQR is primarily involved in sulfide detoxification, can enable the temporary use of sulfide as an alternative electron donor, and may feed electrons into the plastoquinone pool. Nevertheless, we agree that without physiological evidence, this cannot be firmly stated, and we have revised the manuscript accordingly (lines 200 to 209 of the tracked changes document).

Reviewer #2 (Comments to the Author (Required)):

Comment 8: In this revision, the authors have clarified the methods used to infer the early-branching nature of SQR-I in Gloeobacterales. It provides the needed context to understand how the inference was made and how the tree was constructed. Yet, it doesn't address the issues raised. The rooting of Gloeobacterales SQR as ancestral lineages is based on an unrooted tree. While it is true that Gloeobacterales SQR appears to be divergent, this doesn't equate to ancestral relationship. The robustness and node support of the unrooted is also not provided.

We emphasize that our interpretation does not rely on arbitrary rooting, as shown in Supplemental Figure 2 (B). We have revised the manuscript to clarify this important point (lines 139 to 156 of the tracked changes document). Rather, the evolutionary placement of the Gloeobacterales SQR-I lineage is discussed in the context of its phylogenetic position, sequence divergence, and comparative distribution across Cyanobacteria. These analyses were conducted following established phylogenetic best practices routinely applied in our laboratory, which has longstanding expertise in deep evolutionary reconstructions and phylogenomics (Philippe et al., 2011).

The node support values are provided in the zoomed version of Supplementary Figure 2 (B). To further improve transparency, we have now made the complete unrooted tree with all node support values publicly available on Figshare and clearly referenced it in the revised manuscript (Figshare: 10.6084/m9.figshare.31333204).

Comment 9: Most importantly, the inference of the lifestyle of ancestral Cyanobacteria remains to be based only on a single gene and a single tree. There doesn't appear to have sufficient depths nor a sufficient justification for the significant implication of a potential early-branching SQR enzyme in Gloeobacterales.

We would like to clarify that the primary objective of this study is to demonstrate that members of Gloeobacterales do, in fact, encode an SQR contrary to previous assumptions that this lineage lacked the enzyme (Tan et al., 2024: “*the sulfide-driven Anoxygenic Photosynthesis (Anoxygenic Photosynthesis) may not represent an innate feature in Cyanobacteriota. Nevertheless, we cannot rule out the possibility that an as yet unidentified deepbranching lineage may possess Anoxygenic Photosynthesis after the emergence of Cyanobacteriota. More importantly, for Cyanobacteriota living in the Proterozoic oceans, it was crucial to develop this sulfide-driven Anoxygenic Photosynthesis, which couples the consumption of toxic sulfides with energy conservation.*”). Gloeobacterales possess a distinctive characteristic in that they perform oxygenic photosynthesis in the absence of thylakoid membranes, a feature of considerable evolutionary interest. In this context, we also explore why the SQR-I identified in certain Gloeobacterales appears to be associated with lineages detected in cold environments, while carefully avoiding overinterpretation of this ecological pattern.

Importantly, our conclusions regarding the presence and distinctiveness of this SQR are not based solely on a single tree. We performed multiple complementary analyses, including SRA mining to assess environmental distribution (Chikhi et al., 2024) (Figure 1), extensive homology searches within an updated dataset derived from GTDB (Parks et al., 2022) (Supplementary Figures 1 and 2), and protein sequence alignments along with the use of InterProScan (Jones et al., 2014) to evaluate the conservation of key functional regions (Supplementary Figure 3). We have tested multiple evolutionary models (maximum-likelihood under the C20 and LG+C20 models and Bayesian inference under the CAT-Poisson) to ensure that the sequences are consistently recovered as early-diverging. Nevertheless, we acknowledge that a long-branch attraction artifact cannot be entirely excluded. In such a scenario, the interpretation would differ and would instead suggest a more recent horizontal transfer into cold-adapted Gloeobacterales. Nevertheless, this scenario does not appear to be supported by our extensive orthology searches, which did not identify any donors for the putative HGT. These points have now also been clarified in the revised manuscript (lines 19 to 24, 34 to 36, and 139 to 165 of the tracked changes document)

References

- Arieli, B., Shahak, Y., Taglicht, D., Hauska, G., & Padan, E. (1994). Purification and characterization of sulfide-quinone reductase, a novel enzyme driving anoxygenic photosynthesis in *Oscillatoria limnetica*. *Journal of Biological Chemistry*, 269(8), 5705–5711. [https://doi.org/10.1016/S0021-9258\(17\)37518-X](https://doi.org/10.1016/S0021-9258(17)37518-X)
- Cardona, T., Murray, J. W., & Rutherford, A. W. (2015). Origin and Evolution of Water Oxidation before the Last Common Ancestor of the Cyanobacteria. *Molecular Biology and Evolution*, 32(5), 1310–1328. <https://doi.org/10.1093/molbev/msv024>

- Chikhi, R., Raffestin, B., Korobeynikov, A., Robert Edgar, & Artem Barbaian. (2024). Logan: Planetary-Scale Genome Assembly Surveys Life's Diversity. *bioRxiv*.
<https://doi.org/10.1101/2024.07.30.605881>
- Jones, P., Binns, D., Chang, H.-Y., Fraser, M., Li, W., McAnulla, C., McWilliam, H., Maslen, J., Mitchell, A., Nuka, G., Pesseat, S., Quinn, A. F., Sangrador-Vegas, A., Scheremetjew, M., Yong, S.-Y., Lopez, R., & Hunter, S. (2014). InterProScan 5: Genome-scale protein function classification. *Bioinformatics*, 30(9), 1236–1240.
<https://doi.org/10.1093/bioinformatics/btu031>
- Klatt, J. M., Al-Najjar, M. A. A., Yilmaz, P., Lavik, G., De Beer, D., & Polerecky, L. (2015). Anoxygenic Photosynthesis Controls Oxygenic Photosynthesis in a Cyanobacterium from a Sulfidic Spring. *Applied and Environmental Microbiology*, 81(6), 2025–2031. <https://doi.org/10.1128/AEM.03579-14>
- Lumian, J. E., Jungblut, A. D., Dillion, M. L., Hawes, I., Doran, P. T., Mackey, T. J., Dick, G. J., Grettenberger, C. L., & Sumner, D. Y. (2021). Metabolic Capacity of the Antarctic Cyanobacterium *Phormidium pseudopriestleyi* That Sustains Oxygenic Photosynthesis in the Presence of Hydrogen Sulfide. *Genes*, 12(3), 426.
<https://doi.org/10.3390/genes12030426>
- Miller, S. R., & Bebout, B. M. (2004). Variation in Sulfide Tolerance of Photosystem II in Phylogenetically Diverse Cyanobacteria from Sulfidic Habitats. *Applied and Environmental Microbiology*, 70(2), 736–744.
<https://doi.org/10.1128/AEM.70.2.736-744.2004>
- Parks, D. H., Chuvochina, M., Rinke, C., Mussig, A. J., Chaumeil, P.-A., & Hugenholtz, P. (2022). GTDB: An ongoing census of bacterial and archaeal diversity through a phylogenetically consistent, rank normalized and complete genome-based

taxonomy. *Nucleic Acids Research*, 50(D1), D785–D794.

<https://doi.org/10.1093/nar/gkab776>

Philippe H et al. Resolving Difficult Phylogenetic Questions: Why More Sequences Are

Not Enough. *PLOS Biology* 2011;**9**:e1000602.

<https://doi.org/10.1371/journal.pbio.1000602>

Saw, J. H., Cardona, T., & Montejano, G. (2021). Complete Genome Sequencing of a Novel *Gloeobacter* Species from a Waterfall Cave in Mexico. *Genome Biology and Evolution*, 13(12), evab264. <https://doi.org/10.1093/gbe/evab264>

Sheridan, K. J., Duncan, E. J., Eaton-Rye, J. J., & Summerfield, T. C. (2020). The diversity and distribution of D1 proteins in cyanobacteria. *Photosynthesis Research*, 145(2), 111–128. <https://doi.org/10.1007/s11120-020-00762-7>

Tan, S. (2024). Exploring the Origins and Evolution of Oxygenic and Anoxygenic Photosynthesis in Deeply Branched Cyanobacteriota. *Molecular Biology and Evolution*, 41(8). <https://doi.org/10.1093/molbev/msae151>

Re: Spectrum00423-26R1 (**An Early-Diverging SQR Enzyme in Antarctic Gloeobacterales Indicates Sulfide Tolerance in Thylakoid-Lacking Cyanobacteria**)

Dear Dr. Luc Cornet:

Your manuscript has been accepted, and I am forwarding it to the ASM production staff for publication. Your paper will first be checked to make sure all elements meet the technical requirements. ASM staff will contact you if anything needs to be revised before copyediting and production can begin. Otherwise, you will be notified when your proofs are ready to be viewed.

Sincerely,
Blair Steven
Editor
Microbiology Spectrum